# Usefulness of the Sympto-Thermal Method with Standardized Cervical Mucus Assessment (InVivo Method) for Evaluating the Monthly Cycle in Women with Polycystic Ovary Syndrome (PCOS)

**DOI:** 10.3390/healthcare12111108

**Published:** 2024-05-29

**Authors:** Aneta Stachowska, Aleksandra M. Kicińska, Anna Kotulak-Chrząszcz, Anna Babińska

**Affiliations:** 1Department of Physiology, Faculty of Medicine, Medical University of Gdańsk, 80-211 Gdansk, Poland; 2Center for the Treatment of Infertility and Menstrual Cycle Disorders—InVivo Medical Clinic of Gdansk, 80-306 Gdansk, Poland; aleksandra.kicinska@gumed.edu.pl; 3Department of Histology, Faculty of Medicine, Medical University of Gdańsk, 80-211 Gdansk, Poland; anna.kotulak-chrzaszcz@gumed.edu.pl; 4Department of Endocrinology and Internal Medicine, Medical University of Gdańsk, 80-214 Gdansk, Poland; anna.babinska@gumed.edu.pl

**Keywords:** menstrual cycle, fertility awareness methods, fertility biomarkers, cervical mucus, basal body temperature, polycystic ovary syndrome

## Abstract

(1) Background: FABMs (fertility awareness-based methods) are methods that rely on the observation of clinical signs related to fertility found in women, the so-called fertility bioindicators. They can be a valuable tool for diagnosing monthly cycle disorders and infertility, for example, among patients with PCOS (polycystic ovary syndrome). Until now, it has been difficult for women with PCOS to use FABM, due to the difficulty of describing fertility bioindicators and their disorders due to the biology of the syndrome. The new InVivo sympto-thermal method with standardized cervical mucus assessment may provide a valuable diagnostic and therapeutic tool for observing the monthly cycle in this group of women. (2) Methods: The monthly cycle was evaluated in a group of 32 women of reproductive age. A total of 108 monthly cycle observation cards were analyzed: 35 monthly cycle cards were collected from 18 women with PCOS, and 73 monthly cycle cards collected from 14 healthy women. In addition, 32 pairs of macroscopic and microscopic images were evaluated: 17 pairs from the study group (four subjects) and 15 pairs from women in the control group (six subjects). (3) Results: We showed that in the group of patients with PCOS, menstruation was longer (*p* = 0.000814), the number of mucus peaks was statistically higher (*p* = 0.040747), and the interquartile range (IQR) of the duration of the follicular phase (calculated according to the BBT) was significantly higher (8 days) compared to women in the control group. We also observed that among all the women studied, the microscopic image of cervical mucus correlated with the cycle phase described in the observation card, as determined by reference to the BBT chart, provided that it showed the correct features. (4) Conclusions: Systematic maintenance of monthly cycle observation charts using the InVivo method can be an important supplement to the medical history, as it allows for a thorough assessment of, among others, the timing of monthly bleeding, cervical mucus symptoms, BBT changes, and the duration of the follicular and luteal phases among both healthy and PCOS women.

## 1. Introduction

A normal picture of the monthly cycle is considered a sign of a woman’s health, and she should be able to know and monitor it from the beginning of puberty [1]. The function of the ovaries, their ability to ovulate, changes during physiological changes in a woman’s body, such as first menstruation, pregnancy, lactation, or perimenopause. The most common cause of monthly cycle irregularity associated with ovulatory dysfunction is hormonal disorders. These can result from abnormal function of the hypothalamus, pituitary, thyroid, adrenal glands, ovaries, or existing metabolic and immunological disorders [2,3]. Polycystic ovary syndrome (PCOS), which occurs in at least 10% of all women of reproductive age, is among the most common causes of infertility and is associated with a number of metabolic and immune disorders, leading to a deterioration in the quality of health and life of these patients [4,5].

Up to 70% of cases of PCOS are undiagnosed, making it a significant concern for both patients and healthcare professionals [6]. The increase in the prevalence of PCOS appears to be related to the increasing prevalence of metabolic disorders that lead to pre-diabetic conditions and diabetes, resulting from abnormal insulin and glucose metabolism, as well as the scourge of obesity and an excessive supply of simple sugars in the diet [3]. Recent studies have shown that gut microbiota dysbiosis is also involved in the development of PCOS and may exacerbate inflammation and metabolic disorders. Polyphenols and their metabolites have been shown to have anticancer, antibacterial, vasodilatory, and analgesic properties, which significantly alleviate systemic chronic inflammation occurring in women with PCOS [7].

PCOS may clinically manifest as hyperandrogenism (HA), oligoanovulation (OA), and polycystic ovary morphology (PCOM). Women with PCOS are categorized into four phenotypes: HA + OA + PCOM, phenotype-A; HA + OA, phenotype-B; HA + PCOM, phenotype-C; and OA + PCOM, phenotype D [8]. 

The Rotterdam criteria for the diagnosis of PCOS, despite much debate and clinicians’ reservations, remains the most widely and willingly used tool for diagnosing this complex syndrome. The 2023 modified Rotterdam criteria are recommended, according to which PCOS can be diagnosed if any two of the following symptoms are present: (1) clinical or biochemical hyperandrogenism, (2) symptoms of oligo-/lack of ovulation, and (3) polycystic ovarian morphology on ultrasound, excluding other significant abnormalities [9]. Until now, the ultrasound image of polycystic ovaries was understood as the presence of at least 12 follicles with a diameter of 2–9 mm and/or an ovarian volume > 10 mL [10]. Currently, this criterion has been modified and PCOM (polycystic ovarian morphology) can be diagnosed by transvaginal ultrasound with a transducer frequency of ≥8 MHz when ≥20 follicles per ovary are seen in each ovary or the ovarian volume is ≥10 cm^3^ [9].

Approximately 85–90% of women with infrequent menstruation, usually defined as a cycle length of more than 35 days, have a final diagnosis of PCOS [11]. 

The picture of the monthly cycle in women with PCOS includes the following:No or rare ovulation;Prolonged follicular phase and shortened/absent luteal phase;Increased number of days with cervical mucus (consistently occurring cervical mucus resulting from hormonal disorders typical of PCOS);Occurrence of intra-cyclic bleeding/spotting at baseline resulting from, among others, elevated LH levels [2].

Fertility awareness-based methods (FABMs) can be used as a tool for analyzing monthly cycle disorders. This is a group of methods that relies on the observation of physiologically occurring changes in a woman’s body, which can be used as a method of family planning and to evaluate the cycle in patients with menstrual disorders, including PCOS [12,13,14]. Using FABM methods, women observe vaginal bleeding and cervical mucus patterns and/or other bioindicators of fertility, such as basal body temperature (BBT) and cervical cyclic changes (Figure 1) (Table 1).

It is very difficult for women with PCOS to keep track of fertility bioindicators, since the development of cervical mucus—which depends on the growth of granulosa cells in the follicle maturing in the ovary—can be disrupted by abnormal development of these cells, which does not always lead to ovulation [14]. Elevated progesterone levels observed in these women during the pre-ovulatory phase, originating from excessive follicles luteinized prematurely that have not ovulated, correlate with impaired ovulation process and impaired mucus cycle development [17]. An expression of excessive progesterone secretion by ovarian follicles that have not ovulated is the mucus cycles observed by patients, which are not accompanied by a steady increase in BBT [14].

In the study presented, we undertook an analysis of self-completed questionnaires based on the cycles of patients with PCOS compared to a group of healthy women, based on the use of recognized bioindicators of fertility such as BBT and cervical mucus, assessed in relation to a novel cervical mucus pictorial dictionary.

## 2. Materials and Methods

Monthly cycle pattern was evaluated in a group of 32 women of reproductive age (20 to 32 years) who were never pregnant.

A total of 108 monthly cycle observation cards were analyzed: 35 monthly cycle cards collected from 18 women with PCOS (diagnosis based on Rotterdam criteria) diagnosed with menstrual disorders or infertility, and 73 monthly cycle cards collected from 14 healthy women (with regular menstrual cycles and a biphasic thermal curve) who constituted the control group. The average number of cycle cards analyzed from one patient was 3 (IQR: 1–4).

In addition, a group of 10 women included in the study underwent macro and microscopic analysis of cervical mucus images in relation to the monthly cycle chart. A total of 32 pairs of macroscopic and microscopic images were analyzed: 17 pairs from the study group (4 subjects) and 15 pairs from women in the control group (6 subjects).

All of the women were trained by certified instructors in the proper observation of fertility bioindicators using the sympto-thermal method with a standardized evaluation of cervical mucus (InVivo method). 

The description of cervical mucus was based on the standardization developed in the InVivo method, by using a pictorial dictionary and strictly comparing images of the proband’s own secretions with those from the dictionary. The method was developed based on more than 2000 images of vaginal secretions and nearly 430 cycles compared with the clinical situation of women. Thanks to such standardization, it was possible to reliably determine even a few mucus cycles, among patients with PCOS, who may be particularly “lost” in several times the occurrence of genital tract secretions in the first phase of the cycle, despite the absence of ovulation—confirmed by a spike in BBT. 

Statistical analyses were performed using TIBCO Statistica 14.0.1 Software. In order to test for statistically significant differences between the study groups, the Mann–Whitney U test was performed. The level of statistical significance was taken as *p* < 0.05. Due to the relatively small number and continuous nature of the data, a correction for continuity was applied when calculating *p*-values.

### 2.1. Training in Observation of Fertility Bioindicators

Each participant in the study had an initial training meeting with a cycle observation instructor prior to conducting observations. The instructors were individuals who had been trained in the observation of fertility bioindicators by the study supervisor and had been observing their menstrual cycle using the InVivo method for at least six months prior to the start of the investigation. During the study, the patients included in the study remained under the close supervision of the instructors. The entire project was monitored by a doctor who co-designed the study, conducted the training of the major investigator, and who continuously reviewed the investigator’s work and coordinated the entire research process. 

During the first meeting with the instructor, each woman was presented with the same multimedia presentation on the detailed rules for measuring BBT and the standardized way of observing mucus secretions by palpation test and by viewing with simultaneous photography. 

Each patient then began daily self-observation of clinical signs, naturally occurring bioindicators of fertility: basal body temperature (BBT) and cervical mucus. After completing the first cycle card, each woman met again with the cycle observation instructor to check the correctness of the observation and correct errors on the first observation card. The thermal curve was discussed in detail along with disturbed and missing measurements, interpreting the thermal curve according to the rule of 6 lower temperatures preceding ovulation and 3 higher temperatures beginning phase II of the cycle. In addition, photos of the proband’s own vaginal secretions taken independently at home during each day of observation were juxtaposed with photos from the pictorial dictionary and the mucus description on the chart was corrected [15]. Another, third meeting with the cycle observation instructor took place after another two months of independent observation by the women. 

The cycle observation sheets were continuously monitored by the patient’s instructor (constant telephone contact between the patient and instructor). In addition, instructors regularly consulted their patients’ charts with the person conducting the study. A.M.K. served as the chief instructor at the Infertility Treatment Clinic where the study was conducted, and is the creator of the InVivo method.

### 2.2. Basal Body Temperature (BBT)

Basal body temperature was measured just after waking in basal metabolic conditions, in the mouth under, the tongue, with an electronic thermometer to two decimal places (hundredths parts). Each patient self-selected the time at which she took her BBT measurement and continued this time of measurement in subsequent observed cycles. Any deviations from the baseline measurement time were recorded on the observation card and corrected with the cycle observation instructor at the second and third training meetings. In addition, interferences that could have disturbed the BBT measurement, such as colds, various infections, medication, alcohol consumption, climate change, travel, stress, or exhaustion, were noted on the card. Temperatures that were considered disturbed were not taken into account when interpreting the thermal curve [15].

### 2.3. Thermal Curve Interpretation

When interpreting the thermal curve, attention was paid to the BBT spike that occurred after the development of cervical mucus. Six undisturbed temperatures of low-phase temperatures that preceded the BBT spike and three undisturbed high-temperature phase temperatures were then determined. At the same time, the third temperature of the high-temperature phase fulfilled the so-called ovulatory spike condition, i.e., its value was 0.2 °C higher than the highest temperature from the low-temperature phase. The low-temperature phase was separated from the high-temperature phase by a covering line, which was drawn over the highest of the six low-temperature phase temperatures preceding the BBT spike. A sample cycle chart of a PCOS patient containing the thermal curve with its interpretation and the determined cycle phases in relation to BBT measurements is presented below (Figure 2).

### 2.4. Cycle Observation during Menstruation

During menstruation, patients rated, according to the characteristics that were selected and developed in the InVivo method, whether there was bleeding/spotting/dirtying (depending on the amount of blood and color) on a given day and rated its intensity: profuse/moderate/severe. They also described, according to the InVivo instructions, the occurrence of clots, or a gelatinous, clumped mass of blood that could appear during menstruation.

### 2.5. Cervical Mucus

Cervical mucus was observed according to the standards developed in the InVivo method, i.e., under the as similar conditions as possible. To this end, women paid attention to vaginal secretions at each visit to the toilet (before and after micturition and defecation). Secretions from the vestibule of the vagina were collected by wiping with white, smooth, odorless toilet paper to standardize the substrate and unify the conditions for observing mucus morphology. The mucus features were recorded on a specially created InVivo card, which distinguished 3 mucus zones: fertile—blue/less fertile—green/pathological—yellow. In each zone, different mucus characteristics can be selected, which are important for the clinical assessment of the reproductive tract and the body’s hormonal balance. An example of a cycle chart of a patient with PCOS with a description of the characteristics of the observed mucus, the designation of the mucus peaks (three mucus peaks, including one ovulatory) and the cycle phases in relation to the mucus are shown in Figure 2.

The procedure for collecting vaginal secretions was started before urination/defecation by rubbing the vaginal vestibule with paper from front to back. While rubbing, the sensation that was present in the vestibule of the vagina was assessed (wet/slippery/moist/dry/rough). The mucus that remained on the paper was then evaluated. To evaluate the mucus, the discharge taken on the paper was viewed, followed by a palpation test, i.e., the mucus was picked up from the paper, stretched between the fingers, its consistency was evaluated, a macroscopic photo of the observed mucus was taken in good light, and then compared with the photos of cervical mucus included in the pictorial dictionary for a correct description on the chart (a specially created chart with clinically relevant mucus features not included in other FABMs). After urination/defecation, the entire procedure was repeated again. At the end of the day, each sensation occurring during the rubbing of the vaginal vestibule (wet/slippery/moist/dry/rough) and all mucus characteristics that were observed that day were recorded on the cycle observation card (clear/transparent/stretchable/greasy on paper only/white/muddy/mixed/lecky/sticky/dense/papular/creamy/springy/gummy/brown/yellow/greenish/brown/other) [15]. Examples of macroscopic images of cervical mucus taken by the women participating in the study along with their correct descriptions are shown in Figure 3.

Analyzing cycle observation cards in a group of women with PCOS and healthy women, the following parameters were compared:The cycle length.

The first day of the cycle was considered the first day of bleeding or the first day of one or two days of spotting/dirtying that immediately preceded the onset of bleeding. If bleeding was preceded by a minimum of three days with spotting/dirtying then, the prolonged spotting/dirtying was counted as part of the luteal phase of the previous cycle, and only the first day of bleeding was considered the first day of the cycle (thus, luteal phase failure manifested as endometrial failure could be observed).

The length of the bleeding itself.

The duration of monthly bleeding was assessed by counting the days the patient observed bleeding (regardless of the severity).

The length of the entire menstrual period.

Duration of menstruation was counted from the first day of bleeding or the first day of one- or two-day spotting/dirtying that immediately preceded the onset of full bleeding, to the last day of the occurrence of bleeding/spotting/dirtying (regardless of its intensity).

The number of mucus days.

The number of mucous days was assessed by counting days in which any mucous discharge present on the paper was observed. The calculation did not include days in which a mere sensation in the vaginal vestibule during lapping was present without the presence of mucous discharge on the paper being detected.

The number of days without mucus.

The number of days without the presence of mucus was evaluated by counting the days in which the patient did not observe any discharge present on the paper. Days in which a mere sensation in the vaginal vestibule during lapping was present without any mucus discharge being observed on the paper were included in the calculations.

The number of mucus peaks/number of mucus cycles.

The number of mucus peaks in the first phase of the cycle was evaluated, assuming that the mucus peak is the last day of mucus with high fertility characteristics, i.e., stretchy and/or clear, followed by a sudden change in the appearance of cervical mucus.

The day of the ovulation mucus peak.

The day of the ovulatory mucus peak was evaluated, assuming that it was the day of the last mucus peak that occurred around the BBT spike (i.e., 2 days before day 1 of the spike or 2 days after day 1 of the BBT spike).

The length of the follicular phase according to mucus peak

The length of the follicular phase was assessed in relation to the observation of mucus discharge, counting the number of cycle days from the first day of the cycle to the day of the ovulatory mucus peak (including the day of the peak).

The length of the luteal phase according to the mucus peak.

The length of the luteal phase was assessed in relation to the observation of mucus discharge, counting the number of days from the first day after the ovulatory mucus peak to the day before the onset of the next menstrual period.

The day of the BBT spike.

The day of the BBT spike was considered to be the day when the BBT value was higher than the BBT value occurring for the previous 6 days, and the temperatures occurring for consecutive days (at least two consecutive days) were above the covering line. 

One temperature value before the spike (1 lower);One temperature value after the spike (1 higher);The difference of the 1 temperature higher—1 temperature lower (value of the BBT spike).

The difference between the value of the first temperature in the high-temperature phase and the value of the first temperature in the low-temperature phase was evaluated.

The value of the 2 temperature higher;The difference of the 2 temperature higher—1 temperature lower.

The difference between the value of the second temperature in the high-temperature phase and the value of the first temperature in the low-temperature phase was evaluated. 

The difference of the 2 temperature higher—1 temperature higher.

The difference between the value of the second temperature of the high-temperature phase and the value of the first temperature of the high-temperature phase was evaluated. 

The value of the 3 temperature higher;The difference of the 3 temperature higher—1 temperature higher.

The difference between the value of the third temperature of the high-temperature phase and the value of the first temperature of the low-temperature phase was evaluated. 

The difference of the 3 temperature higher—1 temperature higher.

The difference between the value of the third temperature of the high-temperature phase and the value of the first temperature of the high-temperature phase was evaluated. 

The difference of the 3 temperature higher—2 temperature higher.

The difference between the value of the third temperature of the high-temperature phase and the value of the second temperature of the high-temperature phase was evaluated. 

The length of the follicular phase according to the BBT.

The length of the follicular phase was assessed in relation to the BBT chart, counting the number of cycle days from the first day of the cycle (day 1 of menstruation) to the day before the onset of the BBT spike.

The length of the luteal phase according to the BBT.

The length of the luteal phase was calculated from the first day of the high-temperature phase (from the day of the BBT spike) to the day before the onset of the next menstrual period. 

### 2.6. Microscopic Method: Microscopic Preparations of Cervical Mucus

Microscopic smears of cervical mucus were performed by women while observing cervical mucus. After the palpation test/paper viewing, the cervical mucus present on the finger/paper was applied directly to the microscope slide. Each woman was instructed to apply as thin a layer of mucous secretions as possible to a basal slide. The slides were then left in a dry room to dry completely. Prepared microscopic smears of cervical mucus were evaluated using a Delta-Optical IB-100 inverted phase-contrast microscope for the presence of cervical mucus crystallization, comparing the observed image with pictures showing different ways of mucus crystallization included in scientific publications [17,18,19,20,21].

## 3. Results

The median age of the patients participating in the study was 24 years (IQR: 23–27). The control group had a median age of 23 years (IQR: 21–23), while the group of patients with polycystic ovary syndrome (PCOS) was 26.5 years (IQR: 24–29).

Statistically significant differences (*p* < 0.05) in the duration of the entire menstrual period and the number of peaks of cervical mucus were found between the study groups. 

The median cycle length between the groups was not statistically significant. Among healthy women it was 30 (IQR: 29–32), while among PCOS patients it was 33 days (IQR: 28–35).

There were no statistically significant differences between the groups in the duration of monthly bleeding (excluding spotting and dirtying), which averaged 4 days in both the control group (IQR: 4–5) and the study group (IQR: 3–5.5). 

In turn, statistically significant differences were shown in the duration of the entire menstruation, understood as the total number of days of blood discharge from the vagina, in which spotting and dirtying days were taken into account in addition to the duration of bleeding. We showed that in the study group, the average duration of menstruation was 7 days (IQR: 6.76–7.25) compared to the control group, where the average duration of menstruation was shorter at 6 days (IQR: 5.5–6.5). This difference was statistically significant (*p* = 0.000814). 

We did not observe differences between the number of days in the cycle when cervical mucus was present and the number of days when women did not observe mucus. However, we showed that in the group of women with PCOS, the peaks of mucus in the first phase of the cycle were more numerous (*p* = 0.040747). In the group of patients with PCOS, the average number of mucus peaks in the follicular phase was two (IQR: 1.5–2), compared to the number of one in the control group (IQR: 1–1.5). This suggests an abnormal process of ovarian follicle maturation in the group of women with PCOS.

Statistical analysis showed no significant differences between the groups in the duration of the follicular and luteal phases, as well as in the values and differences in height between individual temperatures around the BBT spike. The follicular phase calculated based on the day of the BBT spike in the group of patients with PCOS lasted an average of 18.5 days (IQR: 16.5–23.5), while in the group of healthy women it lasted 16.5 days (IQR: 15–19). Although the *p*-value was 0.156058, signifying the lack of statistical significance between the groups, the interquartile range of the duration of the first phase was significantly higher in the group of patients with PCOS (8 days) compared to healthy women (4 days). The luteal phase, calculated according to the BBT spike in the study group, lasted an average of 12.5 days (IQR: 11–14), while in the group of healthy women it lasted 13 days (IQR: 12–14). The statistical difference in this studied parameter was *p* = 0.883427.

During the analysis, it was shown that in the study group, the day of the BBT spike occurred on average on the 20th day of the cycle (IQR: 18–24), while in the control group, it occurred on 17.75 days of the cycle (IQR: 16–20). Also, a lower value of the per-ovulatory BBT spike (the difference between the value of the first temperature of the higher and first of the lower temperature phases) in the group of women with PCOS was observed, where it reached an average value of 0.2 °C (IQR: 0.15–0.25 °C), compared to 0.25 °C (IQR: 0.2–0.3 °C) in the group of healthy women at *p* = 0.081596 (Table 2) (Figure 4).

An analysis of the microscopic preparations of the mucus was carried out in relation to the macroscopic image of the same mucus recorded on the photograph by the patient, while keeping an observation of vaginal secretions and the monthly cycle card. It was shown that mucus characterized macroscopically by a significant degree of hydration, which was stretchy, transparent and gave a wet/slippery sensation when rubbing the vaginal vestibule with paper (periovulatory, estrogen-dependent mucus), showed normal morphology on microscopic imaging among both healthy women and those with PCOS, crystallizing into forms resembling fern leaves (L-type mucus), taking the forms of parallel crystals (S-type mucus) or radial forms characteristic of P-type mucus. On the other hand, mucus that was springy/gummy, papular/creamy, having abnormal yellow/greenish/brown/other color (exhibiting features from the yellow field on the InVivo method card) showed indeterminate forms of crystallization that are difficult to assess unequivocally. Example pairs showing macroscopic images of cervical mucus taken by women in the control group and their microscopic images are shown in Figure 5, and by women in the study group in Figure 6.

In addition, we showed that the cervical mucus features recorded in the observation card in both groups of women correlated with the macroscopic image (correct recording of mucus features on the card). Additionally, the macroscopic image of the mucus (the image from the photos) correlated with the image visualized by light microscopy. Capturing this correlation between the macroscopic mucus description on the card and the microscopic image confirms the validity of taking macroscopic images of cervical mucus while learning the InVivo method of observation. Such a standardization of the mucus description on the cycle observation card confirms the effectiveness of the training carried out and provides the possibility of correct inference in clinical practice as to the properties of the mucus and the course of the ovarian cycle.

## 4. Discussion

A large group of women are interested in understanding the biology of their own bodies. A thorough analysis of the monthly cycle can also guide a patient’s diagnosis toward the diagnosis of PCOS as the cause of menstrual disorders or infertility.

The most important pathophysiological feature in women with PCOS is insulin resistance (IR) [22]. IR induced by chronic inflammation and hormonal dysfunction originating in adipose tissue, is amplified by the release of pro-inflammatory adipokines [23]. IRis accompanied by oxidative stress and low-grade inflammation that cause abnormal ovarian follicle growth and impaired oocyte maturation with endometrial receptivity dysfunction, which can lead to infertility in PCOS [24,25,26].

The deleterious effects of hyperinsulinemia in women with PCOS are pleiotropic and manifest themselves in different forms at different times in their lives [3,27]. In PCOS, there is central dysregulation of the hypothalamic–pituitary–ovarian (HPO) axis with rapid pulsing of the gonadotropin-releasing hormone GnRH. This is followed by increased pulsation of luteinizing hormone (LH), which stimulates androgen production in the ovaries and interferes with ovulation. Absence or infrequent ovulation generates chronic progesterone deficiency. Inhibitory progesterone feedback, which normally slows down GnRH and LH, is reduced or absent in PCOS [28]; this triggers increased androgen production. Hyperandrogenism contributes to cosmetic defects such as acne and hirsutism, and low-grade inflammation in the vascular endothelium. This leads to impaired lipid metabolism and triggers atherogenic effects leading to the development of metabolic syndrome [29].

The majority of participants in the study conducted by Stujenske et al. also reported a high degree of satisfaction with the use of cycle-tracking technologies and felt that their use contributed to an increased knowledge about reproductive health, indicating the need to improve FABM and technologies for tracking monthly cycle images [30].

Thanks to daily careful observation of fertility bioindicators, women using FABM notice alarming symptoms in the reproductive system earlier and consult a doctor sooner. FABM can also be a valuable diagnostic tool in patients with cycle disorders and infertility and can be used to identify fertile days both by women who are planning a pregnancy and by women who want to postpone conceiving a child. Moreover, it is a cheap method (all you need are the following items: a thermometer, a notebook for recording observations, and a course with a qualified instructor) and does not require pharmacotherapy, so its use is not associated with the risk of side effects. Due to young people’s interest in a natural approach to many areas of life (nutrition, occupational hygiene, and physical activity), this is a method for this group of women. In the context of the infertility treatment and cycle disorders, these methods may be a valuable tool allowing for the proper application of progesterone in luteal failure, in the treatment of PCOS or endometriosis. We hope that many other women with disrupted cycles will benefit from using a standardized method.

Using FABM requires systematic investment of time and effort in observing fertility bioindicators, which may be difficult for active women. Their application may also be difficult due to the changing rules of interpretation depending on the patient’s life situation (nulliparity/postpartum/lactation/post-miscarriage, etc.). In addition, acute or chronic stress, diseases, infections in the urogenital system, travel, abnormal lifestyle, and taking medications may disturb the symptoms of fertility bioindicators. The disadvantage of FABM for couples using them for reproductive purposes is the need for several months of training and close supervision of an instructor in the first phase before couples can independently use FABM as a reliable tool for predicting fertile and infertile days. As an alternative to the use of the FABM, there are various devices and electronic applications that collect data and generate them in graphical form (basal body temperature and mucus cycle chart), allowing the user to interpret the image of fertility bioindicators on her own or with the help of an instructor [31].

FABM method is a dual-indicator method that involves the daily measurement of BBT and the evaluation of cervical mucus. Patients using this method take macroscopic images of their own mucus secretions, which are then evaluated against a pictorial dictionary [15]. This dictionary is an organized and accurately described collection of cervical mucus images. During the use of this method, the woman is accompanied by an instructor, who assists in the correct recording of cervical mucus characteristics and BBT measurements on the cycle observation card. The novelty of the InVivo method lies in the first-ever FABM’s combination of BBT measurements and careful observation of cervical mucus with reference to a pictorial dictionary to avoid ambiguity in describing the morphological features of vaginal discharge [14,15].

When conducting observations during monthly bleeding, it is important to observe in detail the intensity of the bloody discharge and take into account all days of bleeding, spotting and dirtying. Often patients do not take into account the days when they observe moderate/weak spotting or dirtying, which can influence patients to misreport the shortened duration of menstruation to the doctor during the collected history. In the present study, patients used monthly cycle analysis cards on which they marked the days on which they observed the occurrence of bleeding, spotting or dirtying, and rated their intensity.

We showed that the duration of the entire menstrual period was statistically longer in the group of patients with PCOS compared to women in the control group, but the number of days in which bleeding alone occurred (without spotting or menstrual dirtying) was not statistically different in the two groups. This indicates that the endometrial insufficiency found in PCOS is related to progesterone deficiency and lack of stabilization of the endometrium in this group of patients. According to the study, careful observation of the duration of menstruation preceded by training with a qualified instructor allows observation of features of disturbed menstruation that may suggest the diagnosis of PCOS as a source of reproductive health problems.

One of the bioindicators of fertility used in the InVivo method presented in the study is basal body temperature (BBT). The increase in BBT is due to the thermogenic effect of progesterone through its influence on the thermoregulatory center in the hypothalamus [32]. After ovulation, the granulosa cells of the ovarian follicle undergo luteinization and form the so-called corpus luteum in the ovary. At the end of the luteal phase, if conception does not occur, the corpus luteum regresses and the serum progesterone concentration drops sharply, causing the BBT to drop to the values found in the first phase of the cycle. This biphasic BBT pattern retrospectively suggests the onset of ovulation. Daily BBT monitoring is one of the simplest and least invasive methods of monitoring ovulation. BBT is the body temperature that occurs during BMR (basal metabolic rate) and takes place during sleep.

Measurement of BBT does not provide information on the beginning of the fertile period, but only its end, i.e., by changing BBT we can determine the duration of phase I of the cycle and presumed ovulation [33]. Chart of the changes in BBT by indirectly imaging changes in serum progesterone levels during the monthly cycle, can indicate luteal phase abnormalities due to abnormal/insufficient progesterone secretion. Conducting daily BBT measurements can be successfully used in daily clinical practice to properly incorporate this second-phase hormone for pharmacotherapy of luteal insufficiency without disrupting the ovulatory process [14,34].

Analyzing the graphs of thermal curves, we observed that in the group of women with PCOS, the BBT spike occurred later than in the group of healthy women, and the periovulatory BBT spike showed lower values than in the group of healthy women. Although both parameters did not show statistical significance, they suggest abnormalities in the follicle growth phase as well as the ovulation process itself among women with PCOS.

Another bioindicator of fertility used in the InVivo method is cervical mucus, which is a viscous fluid produced by the secretory cells of the cervical crypts. It consists mainly of water, fatty acids, carbohydrates, cholesterol, proteins and inorganic ions, as well as immune system factors that constitute the first defense barrier against pathogenic microorganisms [30,35]. Depending on their location in the cervix, cervical crypts have different receptors for hormones and, depending on the phase of the ovarian cycle, secrete different types of cervical secretions [36]. Cervical mucus produced and stored by the cervical crypts accepts, filters, prepares and stores male sperm deposited in the woman’s vaginal vault during sexual intercourse. In addition, cervical mucus is a barrier preventing pathogens from entering the endometrium through its content of immune factors such as IgA, IgG, cytokines, and antimicrobial proteins [30,37,38,39]. Changes in the composition, quality, and quantity of cervical mucus due to either urogenital microbial carriage or systemic disorders affecting the function of cervical crypts, can result in abnormal functioning of the cervix as an immune and anti-infective valve for the upper floors of the reproductive system [40]. An established researcher who has studied the issue of cervical mucus over the years is Erik Odeblad [20,41].

Outside the periovulatory period, women observe a small amount of thick and sticky mucus, under physiological conditions this occurs outside the ovulatory window. During the periovulatory period, under the influence of an increase in E2 concentration, there is an increase in water production and a decrease in mucin production [33]. As a result, the structure of cervical mucus loosens and it becomes highly susceptible to sperm penetration [41]. During this time, women observe an increased amount of watery mucus discharge that resembles raw egg white in appearance and malleability. The last day of cervical mucus with high fertility characteristics (i.e., extensibility and/or transparency) is called the peak mucus day [33]. An analysis by Richard Fehring showed that 97.8% of mucus peaks are within 4 days of the estimated ovulation day and occur most often on the day of the LH (luteinizing hormone) spike [42].

Cervical crypts produce cervical mucus with different chemical composition in response to changes in estradiol and progesterone concentrations. Taking these changes into account, cervical mucus has been divided into four main mucus types: three estrogenic mucus types (E) and one gestagenic mucus type (G). Each mucus type is produced at a specific point in the cycle and exhibits the ability to crystallize into characteristic patterns/arrangements that depend on the arrangement of the glycoprotein network [16,43] (Table 1) (Figure 1).

Cervical mucus analysis is an important bioindicator of a woman’s monthly cycle. In the physiological monthly cycle, the mucus cycle observed from a woman’s genital tract is continuous, lasting several days uninterruptedly, and is an exponent of the growth of the dominant follicle in the ovary towards ovulation. A normal mucus cycle begins a few days after the end of menstruation (after 2–3 days—these are the so-called dry days, when the follicle is still small and contains a small number of granulosa cells) and reflects the phase of proliferation of granulosa cells in the wall of the growing follicle. The end of the mucus cycle ends the first, follicular phase of the cycle with ovulation and marks the end of the fertile period. A different picture is presented by the cycles of PCOS patients who have intermittent mucus cycles, the so-called mucus patches, resulting in an increased number of mucus peaks in the first phase of the cycle. Mucus showing fertile characteristics alternates with days when mucus is absent. This is due to the presence of numerous follicles undertaking growth towards ovulation, but not completing the growth process [21].

In our study, we confirmed the occurrence of intermittent mucus cycles and, thus, an increased number of mucus peaks in a group of women with PCOS. In a physiological, undisturbed monthly cycle, we expect to see one mucus cycle ending with a BBT spike. In the case of patients with PCOS, we observed the occurrence of an increased number of mucus cycles that were not accompanied by a BBT spike. This confirms that careful observation of the mucus symptom by the patient and systematic maintenance of observation cards can help more quickly target the diagnosis of cycle or menstrual irregularities and infertility to PCOS-related disorders.

Vigili et al. observed differences in the ultrastructure and crystallization of cervical mucus in women with PCOS compared to healthy women. In the control group, the mucus crystallized taking the forms typical of estrogenic mucus: L, P2, S, or P6. In contrast, women with PCOS showed indeterminate forms of mucus crystallization, as well as patches of crystallization resembling estrogenic and pregnancy mucus. The researchers also observed a significant difference in the diameter of the pores present in the cervical mucus mesh, which were 15 µm in women with normal cycles, 8.4 µm in those with PCOS and ovulatory cycles, and 1.8 µm in those with PCOS and non-ovulatory cycles [21]. Cervical mucus is the medium that determines the migration and maturation of sperm in the female reproductive system [20], so any abnormalities that occur in its ultrastructure directly contribute to difficulties in conceiving and, in the long run, infertility, which is one of the most serious consequences of PCOS. Observation of cervical mucus and the ability to determine the peak of the mucus is therefore a very accurate way to determine peak fertility and a fairly accurate way to determine the day of ovulation, as well as the beginning and end of the fertile period [44,45].

Observation of the mucus symptom in conjunction with daily BBT measurement is useful for more accurately identifying the timing of ovulation, especially among patients experiencing several first-phase mucus cycles associated with PCOS. A natural consequence of this syndrome is that, among women with PCOS, determining the correct mucus cycle that ends in ovulation is impossible if not further supported by BBT measurement. Evaluating the timing of the release of the ovum allows the cycle to be divided into two phases: the follicular phase, or the phase of ovarian follicle growth, in which cervical mucus is present and BBT is maintained at a lower level, and the luteal phase, or the corpus luteum phase, in which cervical mucus disappears and BBT reaches higher values. This knowledge allows the woman to know the length of the two phases of her cycle, and the doctor to switch on progesterone to maintain the luteal phase, in a manner tailored to the patient’s individual cycle. This approach allows for precise, personalized treatment tailored to the ovarian cycle currently in progress, according to the actual course of its phases. The pre-ovulatory, follicular phase fluctuates more frequently than the luteal phase [46].

Our study showed that in the group of patients with PCOS, the follicular phase, calculated according to BBT, lasted longer on average compared to patients in the control group. Despite the lack of statistical significance in this parameter between the groups, we observed that the interquartile range of the duration of the first phase in the group of women with PCOS was significantly higher than in the group of healthy women.

Research on FABM to date primarily relates to determining their effectiveness in avoiding pregnancy, the degree of user satisfaction, and comparing different menstrual cycle observation applications [30,47,48]. No studies have yet been published that evaluate the reliability of women’s cycle observations, evaluating the correctness of the examination and documentation of the data from observation cards.

Only a standardized form of notation and observation gives the doctor the opportunity to reliably conclude, especially among patients who experience cycle disturbances, reflected in the quantity and quality of mucus, as well as the course of the basal body temperature curve. Keeping observation cards makes it possible to interpret basal body temperature depending on what is happening in parallel in the mucus cycle. To date, our work has focused on evaluating the usefulness of InVivo cards in the treatment of women with infertility and cycle disorders, including PCOS. The current study is one of several components of a broader project we have undertaken. Our experience over the years indicates that only a proper understanding of the mucus evolution on the card and a proper interpretation of the basal body temperature gives the possibility of proper progesterone supplementation without blocking one’s own ovulation [14].

The most important thing seems to be the standardization of the method and the verification of subjective interpretations of mucus, so that the doctor can correctly conclude and accurately target the diagnosis of menstrual cycle disorders or infertility. For this reason, we proceeded with the current study [18,20,49].

The study had the advantage of parallel observation of the basal body temperature and cervical mucus of the participating women and their regular contact with a menstrual cycle monitoring instructor. A significant benefit of the study was that all women photographed their cervical mucus and compared its appearance to the image of mucus presented in the pictorial dictionary, which minimized the possibility of error in noting mucus characteristics on the observation card. It also provided an opportunity to detect mucus with infectious characteristics and prompted further diagnostics.

A disadvantage of the study is the small size of the study groups and the lack of additional data, e.g., BMI values, fasting glucose and insulin levels, total cholesterol, LDL fraction cholesterol, HDL fraction cholesterol, triglyceride levels, androgen hormone levels, in order to draw attention to the fact that cycle disorders may be a prelude to metabolic disorders occurring in women with PCOS. On the other hand, the study enrolled young women in whom these metabolic disorders may not yet have appeared.

The presence of a statistical age difference between the study groups can also be considered a negative aspect of the presented study. The control group was characterized by a median age of 23 years (IQR: 21–23), while the group of PCOS patients was 26.5 years (IQR: 24–29). Such a phenomenon may be due to the natural history of the clinical course of PCOS. With age, the number of triggers of this immuno–metabolic–hormonal syndrome increases and modulates the function of ovaries prone to the onset of this syndrome. An increasing number of infections, increasing duration of exposure to a diet rich in simple sugars, and increasing visceral fat with age, may be associated with the manifestation of PCOS in increasingly older women. Although women in the control group who were younger may have experienced menstrual cycle irregularity for a variety of reasons not necessarily related to PCOS, statistically significant differences were observed between the study groups. These included differences in the duration of the entire menstrual period and the number of mucus peaks. We observed that the average duration of the follicular phase calculated by BBT, the day of the BBT spike and also the value of the spike itself in the group of women with PCOS differed from those found in the group of healthy women. Although the original assumption regarding the age of women was met—patients aged 20–32 were eligible for the study, when constructing further studies on this topic, attention should be paid to maintaining age homogeneity between the study groups.

## 5. Conclusions

The picture of fertility bioindicators in one cycle can differ radically from that observed in a subsequent cycle in the same woman. The differences between the course of cycles may be due to the influence of endo- and exogenous factors, acting temporarily or chronically, and accompanying acute or chronic diseases. In a group of women with PCOS, which is a heterogeneous clinical entity, we can observe significant differences in cycle patterns both between different patients and in the same woman. For this reason, there is a need for personalized management in the diagnosis and therapy of this group of patients with regard to the differential function of the ovary in each successive cycle. A helpful tool in this regard is the use of the InVivo method of observation of fertility bioindicators, thanks to which a thorough observation of mucous secretions recorded in a standardized manner, according to a pictorial dictionary, minimizes the risk of subjective error in the description of mucous secretions by the patient and allows the doctor to make correct conclusions in the diagnostic and therapeutic process.

The standardization of the InVivo sympto-thermal method through the registration of the macroscopic image of cervical mucus, supported by the education of this group of patients, can be an interesting proposal for women who do not choose to use hormonal contraception or have contraindications to its use, and would like to consciously track the course of their monthly cycles.

The non-standard mucus cycle and abnormalities of both phases of the cycle that occur in PCOS can make it much more difficult for women to correctly diagnose fertility and for physicians to properly initiate chronic therapy with, among others, progesterone routinely used in the condition. The innovative combination of a description of vaginal secretion according to the pictorial dictionary used in the InVivo method with simultaneous measurement of BBT can be a valuable diagnostic tool for monitoring treatment progress among women with cycle disorders, including patients with PCOS.

The study shows that observation of the menstrual cycle is a valuable source of information about a woman’s health. The cycle card can provide a starting point for the diagnosis of menstrual cycle abnormalities and suggest further investigation of, e.g., additional metabolic disorders that occur in a group of women with PCOS.

## Figures and Tables

**Figure 1 healthcare-12-01108-f001:**
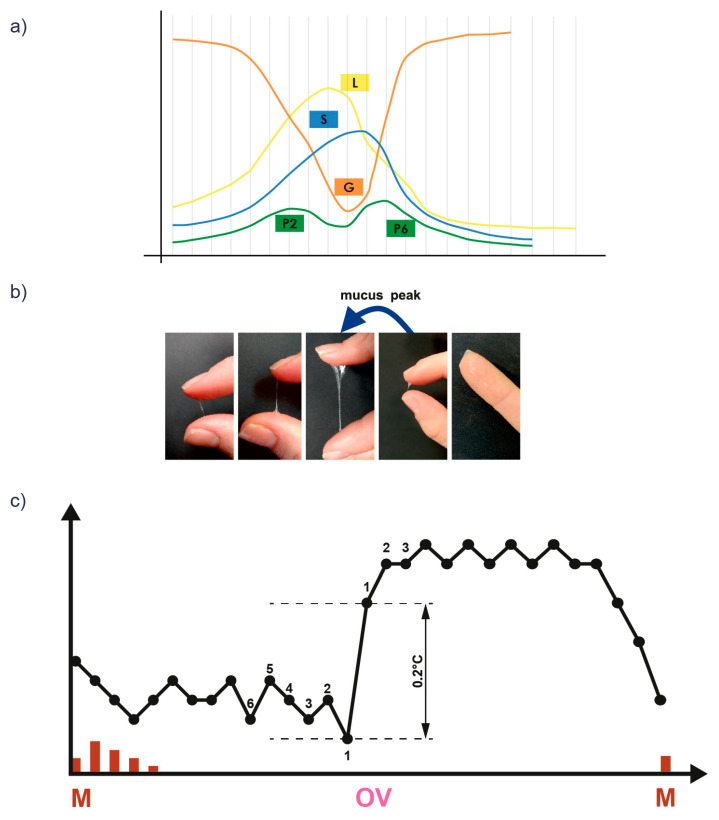
Changes during a woman’s monthly cycle in: (**a**) the number of different mucus types, (**b**) the cervical mucus pattern, and (**c**) BBT values (based on [15,16]).

**Figure 2 healthcare-12-01108-f002:**
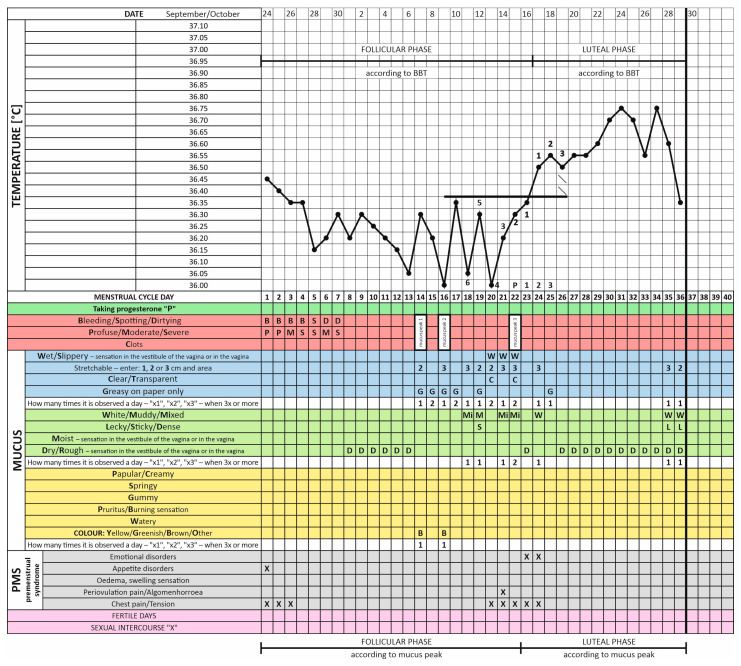
An example of an InVivo cycle observation card filled out by a PCOS patient under the supervision of an instructor, showing an image of the thermal curve (values of individual BBT measurements) and a record of the characteristics of the observed cervical mucus according to the standardization and description contained in the pictorial dictionary of this method. The chart distinguished the division of the cycle into two phases, according to the BBT (an interpretation of the thermal curve was shown, with 6 low temps, 3 high temps and the overlapping line marked) and according to the cervical mucus. Three cervical mucus peaks were marked, with the last one considered appropriate, i.e., periovulatory, as it coincided with the simultaneous occurrence of a BBT spike. Mucus features were divided into three zones: fertile—blue/less fertile—green/pathological—yellow.

**Figure 3 healthcare-12-01108-f003:**
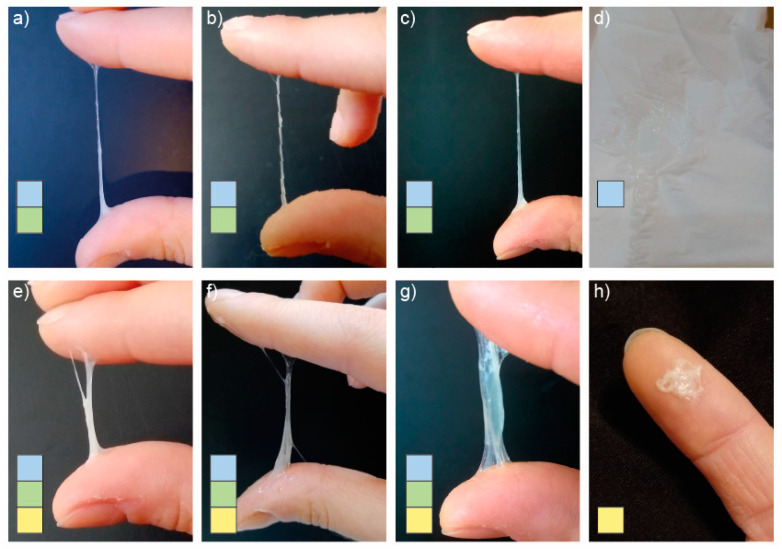
Cervical mucus: (**a**) stretched over 3 cm, clear, mixed (3,C,M)—blue and green field on the card; (**b**) stretched over 3 cm, cloudy (3,C)—blue and green field on the card; (**c**) stretched over 3 cm, mixed (3,M)—blue and green field on the card; (**d**) vitreous only on the paper (V), without the possibility of collecting mucus from the paper—blue field on the card; (**e**) stretched to 2 cm, white, springy (2,W,S)—blue, green and yellow field on the card; (**f**) stretched to 3 cm, mixed, springy, yellow (3,M,S,Y)—blue, green and yellow field on the card; (**g**) stretched to 2 cm, mixed, springy, greenish (2,M,S,G)—blue, green and yellow field on the card; (**h**) papular, yellow (P,Y)—yellow field on the card.

**Figure 4 healthcare-12-01108-f004:**
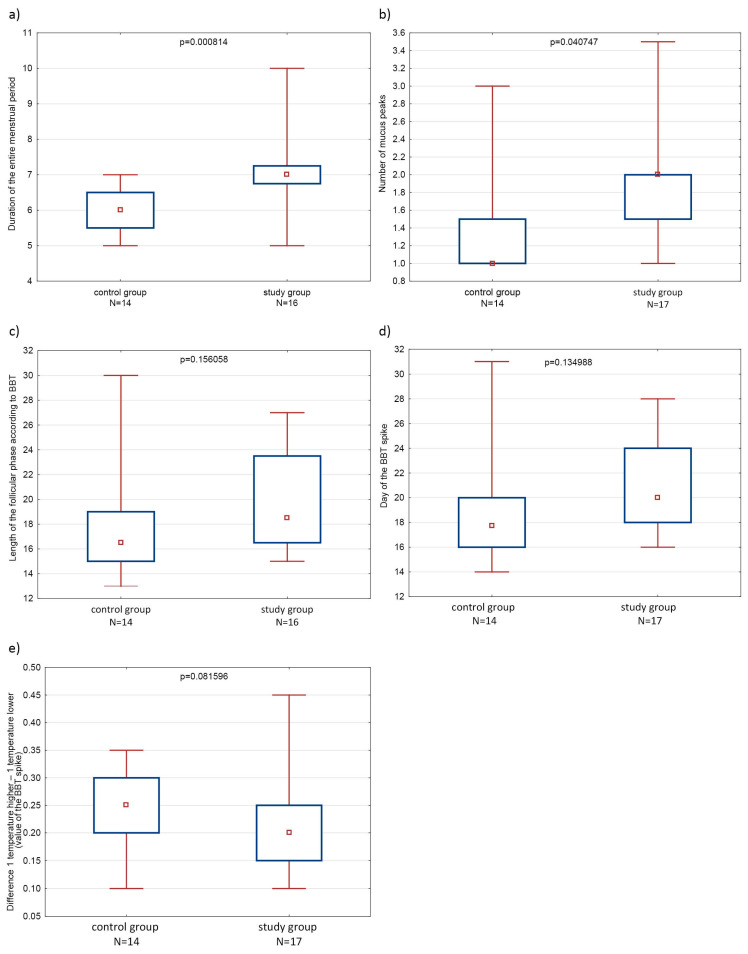
Box-and-whisker plots showing differences between study groups by Mann–Whitney U test in: (**a**) duration of the entire menstrual period (**b**) number of mucus peaks (**c**) length of the follicular phase according to BBT (**d**) day of the BBT spike (**e**) value of the BBT spike; level of statistical significance: *p* < 0.05.

**Figure 5 healthcare-12-01108-f005:**
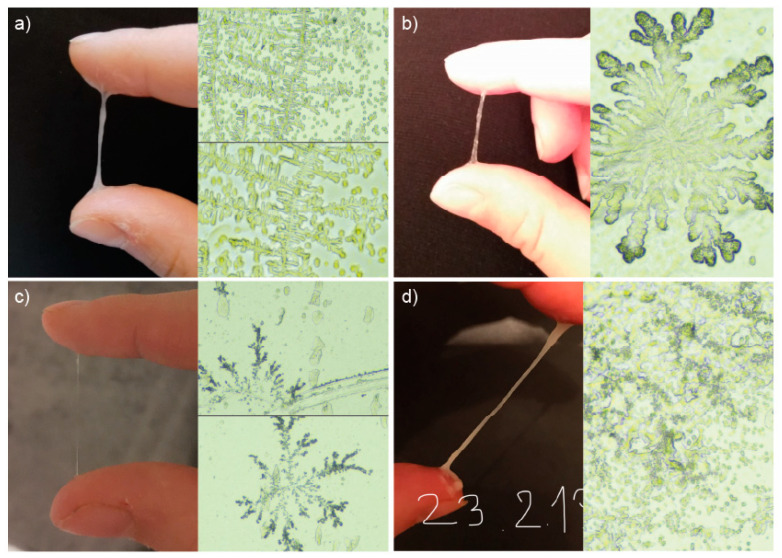
Examples of macroscopic and microscopic images showing mucus observed by women in the control group: (**a**) stretchable and white taken on 17 d.c., 2 days before the mucus peak, showing L-type crystallization (magnification 10× and 20×); (**b**) stretchable and mixed taken on 15 d.c., 3 days before mucus peak, showing P6-type crystallization (magnification 10×); (**c**) stretchable and clear taken 21 d.c., on the day mucus peak, showing Pa- and S-type crystallization (magnification 10×); (**d**) stretchy, cloudy, springy, brown, taken on 19 d.c., on the day of mucus peak, showing fields of crystallization difficult to evaluate (magnification 10×). This may be related to the presence of microorganisms or other inappropriate factors. This type of mucus macroscopically suggests a reproductive tract infection or colonization with abnormal bacterial flora.

**Figure 6 healthcare-12-01108-f006:**
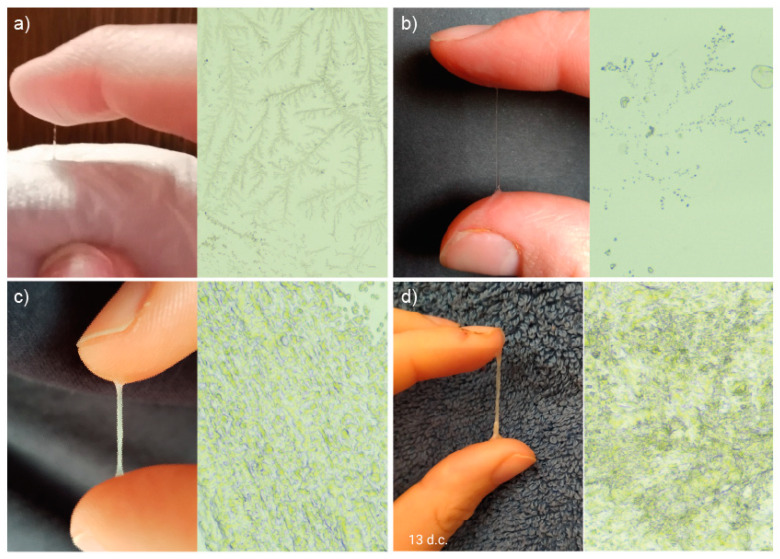
Examples of macroscopic and microscopic images showing the mucus observed by the women in the study group: (**a**) stretchable and clear, taken 8 d.c., on the day of the first mucus peak, showing P2-type crystallization (magnification 4×); (**b**) stretchable and clear, taken 20 d.c., on the day of mucus peak, showing Pt-type crystallization (magnification 10×); (**c**) stretchable, cloudy, yellow, taken 13 d.c., 4 days before mucus peak, showing fields of crystallization difficult to evaluate (magnification 10×); (**d**) stretchable, cloudy, springy, brown, taken 13 d.c., 4 days before mucus peak, showing fields of crystallization difficult to evaluate (magnification 4×).

**Table 1 healthcare-12-01108-t001:** Four main types of cervical mucus.

	Secretion Time	Function
**Estrogenic Mucus Type:**		
L-type mucus (loaf)	secreted under the influence of estrogens 6–7 days before ovulation	together with S mucus, prevents abnormal sperm from passing through
S-type mucus (string)	secreted under the influence of estrogens 2–3 days before ovulation	is responsible for transporting normal sperm, along with mucus L, prevents abnormal sperm from passing through
P-type mucus (peak)	secreted under the influence of estrogen, especially on the peak day, i.e., the day of ovulation and the following day	transports sperm from S crypts to the uterine cavity and exhibits mucolytic activity
**Gestagenic mucus type:**		
G-type mucus	is secreted under gestagenic conditions, during infertile periods of the cycle	prevents sperm from passing through

**Table 2 healthcare-12-01108-t002:** Results of the analysis of individual parameters assessed by the monthly cycle analysis questionnaires in the control and study groups. The table shows the number of women studied, median, lower and upper quartile values, and *p*-values for the Mann–Whitney nonparametric U test; the level of statistical significance: *p* < 0.05.

	Control Group				Study Group				*p*
	N	Median	Lower Quartile	Upper Quartile	N	Median	Lower Quartile	Upper Quartile	
Age	13	23.00	21.00	23.00	18	26.50	24.00	29.00	0.003300
Number of cycles analyzed	14	3.50	3.00	4.00	18	1.50	1.00	3.00	0.000420
Cycle length	14	30.00	29.00	32.00	17	33.00	28.00	35.00	0.339053
Length of bleeding itself	14	4.00	4.00	5.00	16	4.00	3.00	5.50	0.733841
Length of entire menstrual period	14	6.00	5.50	6.50	16	7.00	6.75	7.25	0.000814
Number of mucus days	14	13.75	11.00	20.00	15	17.00	9.00	19.00	1.000000
Number of days without mucus	14	9.00	4.00	12.00	14	7.50	4.00	14.00	0.871907
Number of mucus peaks	14	1.00	1.00	1.50	17	2.00	1.50	2.00	0.040747
Day of ovulation mucus peak	14	17.50	15.00	19.00	17	19.00	17.00	22.00	0.209875
Length of follicular phase according to mucus peak	14	17.50	15.00	19.00	17	19.00	17.00	22.00	0.209875
Length of luteal phase according to mucus peak	14	12.00	11.00	13.00	16	12.00	11.50	14.00	0.966289
Day of BBT spike	14	17.75	16.00	20.00	17	20.00	18.00	24.00	0.134988
Value of 1 temperature before spike (1 lower)	14	36.2625	36.15	36.30	17	36.30	36.20	36.40	0.139018
Value of 1 temperature after spike (1 higher)	14	36.475	36.40	36.525	17	36.50	36.40	36.60	0.424589
Difference 1 temperature higher—1 temperature lower	14	0.25	0.20	0.30	17	0.20	0.15	0.25	0.081596
Value of 2 temperature higher	14	36.4875	36.40	36.60	17	36.55	36.40	36.60	0.734576
Difference 2 temperature higher—1 temperature lower	14	0.30	0.25	0.35	17	0.30	0.175	0.35	0.734211
Difference 2 temperature higher—1 temperature higher	14	0.0625	0.00	0.10	17	0.05	0.00	0.10	0.780364
Value of 3 temperature higher	13	36.50	36.45	36.625	16	36.625	36.4875	36.70	0.084687
Difference 3 temperature higher—1 temperature higher	13	0.30	0.25	0.35	16	0.3125	0.25	0.375	0.480840
Difference 3 temperature higher—1 temperature higher	13	0.05	0.00	0.10	16	0.1125	0.05	0.175	0.172680
Difference 3 temperature higher—2 temperature higher	13	0.00	0.00	0.10	16	0.05	−0.025	0.15	0.414084
Length of follicular phase according to BBT	14	16.50	15.00	19.00	16	18.50	16.50	23.50	0.156058
Length of luteal phase according to BBT	14	13.00	12.00	14.00	16	12.50	11.00	14.00	0.883427

## Data Availability

Data supporting the reported results can be obtained from the corresponding author upon any reasonable request.

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
