# Peer review of "Usefulness of the Sympto-Thermal Method with Standardized Cervical Mucus Assessment (InVivo Method) for Evaluating the Monthly Cycle in Women with Polycystic Ovary Syndrome (PCOS)"

_healthcare, 2024, doi:10.3390/healthcare12111108_

Round 1

Reviewer 1 Report

Comments and Suggestions for Authors

I am writing to express my opinion regarding the revised manuscript titled " Usefulness of the sympto-thermal method with standardized cervical mucus assessment (InVivo method) for evaluating the monthly cycle in women with polycystic ovary syndrome (PCOS) ", which was recently submitted to healthcare. Having thoroughly reviewed the manuscript, I would like to share my thoughts on its unsuitability for publication until major modifications

The introduction should focused on the sympto-thermal method?

The introduction is too long and is not well-organized.

What is the gap of knowledge?

What is previous studies findings about this issue?

How calculated the sample size?

What is the inclusion and exclusion criteria?

What was the characteristics of instructors?

Who controlled the bias and errors of reporting the findings by women?

The discussion section should be organized based on the findings, mainly discuss by comparing the other studies.

What is the clinical implication of this study?

Comments on the Quality of English Language

minor edit should be performed.

Author Response

Response to Reviewers comments

Thank you for the opportunity to address the comments from the Reviewers. The authors hope that the Reviewers and Editors will be satisfied with the further amendments which we have made to the manuscript after taking on board the feedback.

Sincerely yours,

Aneta Stachowska

Department of Physiology

Medical University of Gdańsk

+48 58 349 1500

Dębinki 1, 80-211 Gdańsk

Title: Usefulness of the sympto-thermal method with standardized cervical mucus assessment (InVivo method) for evaluating the monthly cycle in women with Polycystic Ovary Syndrome (PCOS)

Reference: 2988683

Answers to the Reviewer 1

Dear Reviewer 1, we are thankful for your insightful comments which helped to, hopefully, improve our manuscript.

The introduction should focused on the sympto-thermal method?

The content of the introduction has been modified, focusing on menstrual cycle disorders in PCOS and presenting basic knowledge about this disorder.

The introduction is too long and is not well-organized.

Based on your comments, the introduction has been shortened and reorganized.

What is the gap of knowledge?

A dual-indicator FABMs (Fertility Awareness-Based Methods) has not yet been developed to serve as a primary source for monitoring the clinical status of a patient's menstrual cycle for diagnostic and personalized therapy as well as the possibility of monitoring the applied treatment for infertility and cycle disorders. The InVivo method is the first tool of its kind. For this reason, women with PCOS, who particularly often experience abnormal cycles and menstruation, were included in our study as the group that could potentially benefit the most from the use of dual-indicator FABMs with the ability to objectify and note the type of cervical mucus according to the pictorial dictionary we introduced. The InVivo method is an proprietary project based on years of experience and work with menstrual cycle analysis. Our study has been approved by the Independent Bioethics Committee for Scientific Research at the Medical University of Gdansk (Resolution No. NKBBN/496/2018-2019) –  the document has been delivered to the Assistant Editor.

What is previous studies findings about this issue?

No study addressing the issue of menstrual cycle observation in a group of women with PCOS was found in the available literature, which demonstrates the innovation of our investigation. As a prelude to the study we created an InVivo observation card, which was published in a case report in the Healthcare. The example of the PCOS patient described in the aforementioned manuscript shows that procreative success was achieved by careful observation of the menstrual cycle and proper application of progesterone in relation to fertility bioindicators [1].

[1.] Kicińska, A.M.; Stachowska, A.; Kajdy, A.; Wierzba, T.H.; Maksym, R.B. Successful Implementation of Menstrual Cycle Biomarkers in the Treatment of Infertility in Polycystic Ovary Syndrome—Case Report. Healthcare 2023, 11(4), 616; https://doi.org/10.3390/healthcare11040616

How calculated the sample size?

We conducted a pilot study, and the sample size was determined based on available resources, such as the duration of the study and the availability of participants (the number of patients presenting to the Infertility Treatment Clinic during the study period and the number of healthy women who were willing to learn how to conduct cycle observations). We plan to continue our study to achieve greater representativeness in the future.

What is the inclusion and exclusion criteria?

The study enrolled women aged 20-32 who had never been pregnant. Women suffered from cycle disorders or infertility diagnosed with PCOS (diagnosis based on the Rotterdam criteria) were recruited to the study group, while women with regular monthly cycles with a biphasic thermal curve were recruited to the control group, as described in the 112-118 line. Exclusion criteria for control group: abnormalities in menstrual cycle regularity and symptoms that may indicate the presence of PCOS or other reproductive health disorders identified by medical history.

What was the characteristics of instructors?

The instructors were individuals who had been trained in the observation of fertility bio-indicators by the study supervisor and had been observing their menstrual cycle using the InVivo method for at least six months prior to the start of the investigation. During the study, the patients included in the study remained under the close supervision of the instructors. The entire project was monitored by a doctor who co-designed the study, conducted the training of the major investigator, and who continuously reviewed the investigator's work and coordinated the entire research process (Aleksandra Maria Kicińska, Ph.D.), as described in the 141-148 line.

Who controlled the bias and errors of reporting the findings by women?

The cycle observation sheets were continuously monitored by the patient's instructor (constant telephone contact between the patient and instructor). Moreover, the instructors consulted their patients' cards on an ongoing basis with the person coordinating the study and being the Chief Instructor at the Infertility Treatment Clinic where the study was conducted (Aleksandra Maria Kicińska, Ph.D.), as described in 165-170 line. Supervision of the entire study and training of instructors was provided by the creator of the InVivo method, a doctor who has been treating women with infertility and menstrual cycle disorders according to fertility bio-indicators for more than 17 years. In addition, this doctor is certified as a Fertility Awareness-Based Methods Instructor by the Institute of Maternal and Child Health (Warsaw, Poland) as well as NFP MD (Natural Family Planning Medical Doctor), and she has more than 20 years of experience teaching women to observe their menstrual cycle in various health situations.

The discussion section should be organized based on the findings, mainly discuss by comparing the other studies.

Research on Fertility Awareness-Based Methods to date primarily relates to determining their effectiveness in avoiding pregnancy, the degree of user satisfaction, and comparing different menstrual cycle observation applications [1] [2] [3]. No studies have yet been published that evaluate the reliability of women's cycle observations, evaluating the correctness of the examination and documentation of the data from an observation cards.

Only a standardized form of notation and observation gives the doctor the opportunity to reliably conclude, especially among patients who experience cycle disturbances, reflected in the quantity and quality of mucus as well as the course of the basal body temperature curve. Keeping observation cards makes it possible to interpret basal body temperature depending on what is happening in parallel in the mucus cycle. To date, our work has focused on evaluating the usefulness of InVivo cards in the treatment of women with infertility and cycle disorders, including PCOS. The current study is one of several components of a broader project we have undertaken. Our experience over the years indicates that only a proper understanding of the mucus evolution on the card and a proper interpretation of the basal body temperature gives the possibility of proper progesterone supplementation without blocking one's own ovulation. We described such a case of a patient in the Healthcare journal [4].

The most important thing seems to be the standardization of the method and the verification of subjective interpretations of mucus, so that the doctor can correctly conclude and accurately target the diagnosis of menstrual cycle disorders or infertility. For this reason, we proceeded with the current study. The inspiration for such a clinical study came from the previously published work of Prof. Erik Odeblad, a Swedish gynecologist who studied the properties of cervical mucus in magnetic resonance imaging (MRI) [5] [6] [7]. 

The discussion has been revised. The current findings are presented and compared with the results of other studies.

  1. Stanford, J.B.; Smith, K.R.; Varner, M.W. Impact of Instruction in the Creighton Model Fertilitycare System on Time to Pregnancy in Couples of Proven Fecundity: Results of a Randomised Trial. Paediatr. Perinat. Epidemiol. 2014, 28, 391–399, doi:10.1111/ppe.12141.
  2. Unseld, M.; Rötzer, E.; Weigl, R.; Masel, E.K.; Manhart, M.D. Use of Natural Family Planning (NFP) and Its Effect on Couple Relationships and Sexual Satisfaction: A Multi-Country Survey of NFP Users from US and Europe. Front. Public Heal. 2017, 5, doi:10.3389/FPUBH.2017.00042.
  3. Stujenske, T.M..; Mu, Q..; Pérez Capotosto, M..; Bouchard, T.P. Survey Analysis of Quantitative and Qualitative Menstrual Cycle Tracking Technologies. Medicina (B. Aires). 2023, 59, 1–10, doi:https://doi.org/10.3390/ medicina59091509.

[4] Kicińska, A.M.; Stachowska, A.; Kajdy, A.; Wierzba, T.H.; Maksym, R.B. Successful Implementation of Menstrual Cycle Biomarkers in the Treatment of Infertility in Polycystic Ovary Syndrome—Case Report. Healthc. 2023, 11, doi:10.3390/healthcare11040616.

[5] Menárguez, M.; Pastor, L.M.; Odebad, E. Morphological Characterization of Different Human Cervical Mucus Types Using Light and Scanning Electron Microscopy. Hum. Reprod. 2003, 18, 1782–1789, doi:10.1093/humrep/deg382.

[6] Odeblad, E. The Discovery of Different Types of Cervical Mucus and the Billings Ovulation Method. Bull. Ovul. Method Res. Ref. Cent. Aust. 1994, 21, 3–35.

[7] Harvey, C.; Linn, R.A.; Jackson, M.H. Certain Characteristics of Cervical Mucus in Relation to the Menstrual Cycle. Obstet. Gynecol. Surv. 1961, 16, 98–100, doi:10.1097/00006254-196102000-00032.

What is the clinical implication of this study?

The study shows that observation of the menstrual cycle is a valuable source of information about a woman's health. The cycle card can provide a starting point for the diagnosis of menstrual cycle abnormalities and suggest further investigation of, e.g., additional metabolic disorders that occur in a group of women with PCOS. A paragraph regarding the clinical implications of the study has been added to the manuscript (lines: 709-712).

Minor edit should be performed.

We have sent the Assistant Editor a certificate of translation into English.

Reviewer 2 Report

Comments and Suggestions for Authors

The article has many good points that certainly deserve to be published but in this form it is too difficult to read and too much.

Figure 6 in the  discussion  is an example of a normal symptom-thermal method. This can best be given in the introduction so that people immediately know what you mean. The same with the type of mucus given in table 2. And make it as short as possible.

Then start your study as described from line 135 untill line 443. And if possible try to shorten this part

What is confusing is the numbering of the pages , there is a 1 to 23 and then a new 1 to 23??? So I used the numbers of the lines.

As said there are many good and interesting parts, but make it shorter. 

Author Response

Response to Reviewers comments

Thank you for the opportunity to address the comments from the Reviewers. The authors hope that the Reviewers and Editors will be satisfied with the further amendments which we have made to the manuscript after taking on board the feedback.

Sincerely yours,

Aneta Stachowska

Department of Physiology

Medical University of Gdańsk

+48 58 349 1500

Dębinki 1, 80-211 Gdańsk

Title: Usefulness of the sympto-thermal method with standardized cervical mucus assessment (InVivo method) for evaluating the monthly cycle in women with Polycystic Ovary Syndrome (PCOS)

Reference: 2988683

Answers to the Reviewer 2

We would like to thank the Reviewer 2 for a careful and thorough reading of this manuscript. We hope that the final version of the manuscript will be satisfactory.

The article has many good points that certainly deserve to be published but in this form it is too difficult to read and too much.

The introduction and discussion have been significantly shortened and reorganized.

Figure 6 in the  discussion  is an example of a normal symptom-thermal method. This can best be given in the introduction so that people immediately know what you mean. The same with the type of mucus given in table 2. And make it as short as possible.

According to the suggestion, a reference to the figure presenting sympto-thermal method and a table explaining different types of mucus were included in the introduction (lines: 90).

Then start your study as described from line 135 untill line 443. And if possible try to shorten this part.

According to the comment, discussion has been reorganized.

What is confusing is the numbering of the pages , there is a 1 to 23 and then a new 1 to 23??? So I used the numbers of the lines.

Page numbering has been changed.

As said there are many good and interesting parts, but make it shorter.

The work has been shortened in accordance with the reviewer's comments.

Reviewer 3 Report

Comments and Suggestions for Authors

Comments

1. Define PCOS using the following citation (https://www.frontiersin.org/journals/endocrinology/articles/10.3389/fendo.2023.1303747/full)

2. Define the prevalence of PCOS in the Poland women.

3. Link between hormonal disorder and PCOS in detail.

4. Is there any other alternative tools apart from FABMs towards analyzing monthly cycle disorders?

5. Define pros and cons towards using FABMs tool.

6. Authors need to describe clearly as how many cases and controls were included in the methodology section?

7. Define inclusion and exclusion criteria of women involved in this study.

8. Authors can alternative cite for the sub-sections from 2.1 to 2.6 sections in the methodology.

9. Authors need to explain or give the reference for including the cases and controls or add the sample size formula.

10. BMI is very important element towards the PCOS women. If there is any probability, authors can add the BMI details in the revised manuscript.

11. Please do add the mean ages of both the groups of this studied women.

12. The majority of the results showed the negative association and it may be due to low sample size. Is there any alternative answer to justify the raised question.

13. The quality of figure-3 is not maintaining the good and high resolution.

14. Authors need to describe and explain in detailed towards using the FABMs method in previous studies and compare the current study results with the prior studies.

15. Can authors explain the impact of this study in PCOS women.

16. Define pros and cons of this study.

17. Precise the conclusion.

Author Response

Response to Reviewers comments

Thank you for the opportunity to address the comments from the Reviewers. The authors hope that the Reviewers and Editors will be satisfied with the further amendments which we have made to the manuscript after taking on board the feedback.

Sincerely yours,

Aneta Stachowska

Department of Physiology

Medical University of Gdańsk

+48 58 349 1500

Dębinki 1, 80-211 Gdańsk

Title: Usefulness of the sympto-thermal method with standardized cervical mucus assessment (InVivo method) for evaluating the monthly cycle in women with Polycystic Ovary Syndrome (PCOS)

Reference: 2988683

Answers to the Reviewer 3

We would like to thank the reviewer for all his valuable comments on the manuscript. We hope that the final version of the manuscript will be satisfactory.

  1. Define PCOS using the following citation (https://www.frontiersin.org/journals/endocrinology/articles/10.3389/fendo.2023.1303747/full)

As suggested by the reviewer, a definition of PCOS was added in the introduction (lines: 60-63) along with the above-mentioned reference.

“PCOS may clinically manifest as hyperandrogenism (HA), oligoanovulation (OA), and polycystic ovary morphology (PCOM). Women with PCOS are categorized into four phenotypes: HA+OA+PCOM, phenotype-A; HA+OA, phenotype-B; HA+PCOM, phenotype-C; and OA+PCOM, phenotype D.”

  1. Define the prevalence of PCOS in the Poland women.

Prevalence of polycystic ovary syndrome in 2016 in Poland for women aged 15–49 years was 447.22  per 100,000 [1], while worldwide this disorder affects 4–20% of women of reproductive age [2].

  1. Miazgowski, T.; Martopullo, I.; Widecka, J.; Miazgowski, B.; Brodowska, A. National and Regional Trends in the Prevalence of Polycystic Ovary Syndrome since 1990 within Europe: The Modeled Estimates from the Global Burden of Disease Study 2016. Arch. Med. Sci. 2021, 17, 343–351, doi:10.5114/aoms.2019.87112.
  2. Deswal, R.; Narwal, V.; Dang, A.; Pundir, C.S. The Prevalence of Polycystic Ovary Syndrome: A Brief Systematic Review. J. Hum. Reprod. Sci. 2020, 13, 261–271, doi:10.4103/jhrs.JHRS_95_18.
  3. Link between hormonal disorder and PCOS in detail.

The most important pathophysiological feature in women with PCOS is insulin resistance (IR) [1]. IR induced by chronic inflammation and hormonal dysfunction originating in adipose tissue, is amplified by the release of pro-inflammatory adipokines [2]. IRis accompanied by oxidative stress and low-grade inflammation that cause abnormal ovarian follicle growth and impaired oocyte maturation with endometrial receptivity dysfunction, which can lead to infertility in PCOS [3][4][5].

The deleterious effects of hyperinsulinemia in women with PCOS are pleiotropic and manifest themselves in different forms at different times in their lives [6][7]. In PCOS, there is central dysregulation of the hypothalamic-pituitary-ovarian (HPO) axis with rapid pulsing of the gonadotropin-releasing hormone GnRH. This is followed by increased pulsation of luteinizing hormone (LH), which stimulates androgen production in the ovaries and interferes with ovulation. Absence or infrequent ovulation generates chronic progesterone deficiency. Inhibitory progesterone feedback, which normally slows down GnRH and LH, is reduced or absent in PCOS [8]. This triggers increased androgen production. Hyperandrogenism contributes to cosmetic defects such as acne and hirsutism, and low-grade inflammation in the vascular endothelium. This leads to impaired lipid metabolism and triggers atherogenic effects leading to the development of metabolic syndrome [9] (lines: 442-459).

  1. Xing, C.; Li, C.; He, B. Insulin Sensitizers for Improving the Endocrine and Metabolic Profile in Overweight Women with PCOS. J. Clin. Endocrinol. Metab. 2020, 105, 2950–2963, doi:10.1210/clinem/dgaa337.
  2. Chen, P.; Jia, R.; Liu, Y.; Cao, M.; Zhou, L.; Zhao, Z. Progress of Adipokines in the Female Reproductive System: A Focus on Polycystic Ovary Syndrome. Front. Endocrinol. (Lausanne). 2022, 13, 1–14, doi:10.3389/fendo.2022.881684.
  3. Calcaterra, V.; Verduci, E.; Cena, H.; Magenes, V.C.; Todisco, C.F.; Tenuta, E.; Gregorio, C.; De Giuseppe, R.; Bosetti, A.; Di Profio, E.; et al. Polycystic Ovary Syndrome in Insulin‐resistant Adolescents with Obesity: The Role of Nutrition Therapy and Food Supplements as a Strategy to Protect Fertility. Nutrients 2021, 13, 1–32, doi:10.3390/nu13061848.
  4. Zuo, M.; Liao, G.; Zhang, W.; Xu, D.; Lu, J.; Tang, M.; Yan, Y.; Hong, C.; Wang, Y. Effects of Exogenous Adiponectin Supplementation in Early Pregnant PCOS Mice on the Metabolic Syndrome of Adult Female Offspring. J. Ovarian Res. 2021, 14, 1–12, doi:10.1186/s13048-020-00755-z.
  5. Lorzadeh, N.; Kazemirad, N.; Kazemirad, Y. Human Immunodeficiency: Extragonadal Comorbidities of Infertility in Women. Immunity, Inflamm. Dis. 2020, 8, 447–457, doi:10.1002/iid3.327.
  6. Barber, T.M.; Franks, S. Obesity and Polycystic Ovary Syndrome. Clin. Endocrinol. (Oxf). 2021, 95, 531–541, doi:10.1111/cen.14421.
  7. Kicińska, A.M.; Maksym, R.B.; Zabielska-Kaczorowska, M.A.; Stachowska, A.; Babińska, A. Immunological and Metabolic Causes of Infertility in Polycystic Ovary Syndrome. Biomedicines 2023, 11, 1567, doi:10.3390/biomedicines11061567.
  8. Shirin, S.; Murray, F.; Goshtasebi, A.; Kalidasan, D.; Prior, J.C. Cyclic Progesterone Therapy in Androgenic Polycystic Ovary Syndrome (PCOS)—A 6-Month Pilot Study of a Single Woman’s Experience Changes. Medicina (B. Aires). 2021, 57, 1024, doi:10.3390/medicina57101024.
  9. Purwar, A.; Nagpure, S. Insulin Resistance in Polycystic Ovarian Syndrome. Cureus 2022, 14, doi:10.7759/cureus.30351.
  10. Is there any other alternative tools apart from FABMs towards analyzing monthly cycle disorders?

Alternative tools include ultrasound, routinely performed in daily gynecological practice, ovulation monitoring, and regular blood hormone concentration testing. However, these methods are often an organizational, economic and often psychological/social barrier. Simpler to perform, and not requiring the involvement of medical staff, are home ovulation tests that examine urinary LH levels. However, among patients with PCOS, these tests are unreliable, due to excessive LH secretion unrelated to ovulation.

As an alternative to the use of the FABM, there are various devices and electronic applications that collect data and generate them in graphical form (basal body temperature and mucus cycle chart) allowing the user to interpret the image of fertility bio-indicators on her own or with the help of an instructor. However, they are not entirely reliable, because one must always take into account incorrect assumptions of the calculations often used by artificial intelligence, which tries to determine a certain model. However, we know that clinically, in relation to menstrual cycles, it is impossible to predict how the body will behave in a given cycle after unplanned exposure to new factors.

The manuscript was supplemented with the above information in the lines 485-489.

  1. Define pros and cons towards using FABMs tool.

Thanks to daily careful observation of fertility bioindicators, women using FABM (Fertility Awareness-Based Methods) notice alarming symptoms in the reproductive system earlier and consult a doctor sooner. FABM can also be a valuable diagnostic tool in patients with cycle disorders and infertility, and can be used to identify fertile days both by women who are planning a pregnancy and by women who want to postpone conceiving a child. Moreover, it is a cheap method (all you need is: a thermometer, a notebook for recording observations and a course with a qualified instructor) and does not require pharmacotherapy, so its use is not associated with the risk of side effects. Due to young people's interest in a natural approach to many areas of life (nutrition, occupational hygiene, physical activity), this is a method for this group of women. In the context of the infertility treatment and cycle disorders, these methods may be a valuable tool allowing for the proper application of progesterone in luteal failure, in the treatment of PCOS or endometriosis, but also in the treatment of PMS – premenstrual syndrome.

Using FABM requires systematic investment of time and effort in observing fertility bioindicators, which may be difficult for active women. Their application may also be difficult due to the changing rules of interpretation depending on the patient's life situation (nulliparity /postpartum /lactation /postmiscarriage, etc.). In addition, acute or chronic stress, diseases, infections in the urogenital system, travel, abnormal lifestyle and taking medications may disturb the symptoms of fertility bioindicators. The disadvantage of FABM for couples using them for reproductive purposes is the need for several months of training and close supervision of an instructor in the first phase before couples can independently use FABM as a reliable tool for predicting fertile and infertile days.

Above information has been added to the discussion (lines: 464-485).

  1. Authors need to describe clearly as how many cases and controls were included in the methodology section?

The methodology section included 32 women: 18 from the study group and 14 from the control group, as described in the lines: 112-118.

“Monthly cycle pattern was evaluated in a group of 32 women of reproductive age (20 to 32 years) who were never pregnant. 108 monthly cycle observation cards were analyzed: 35 monthly cycle cards collected from 18 women with PCOS (diagnosis based on Rotterdam criteria) diagnosed with menstrual disorders or infertility, and 73 monthly cycle cards collected from 14 healthy women (with regular menstrual cycles and a biphasic thermal curve) who constituted the control group.”

  1. Define inclusion and exclusion criteria of women involved in this study.

The study enrolled women aged 20-32 who had never been pregnant. Women suffered from cycle disorders or infertility diagnosed with PCOS (diagnosis based on the Rotterdam criteria) were recruited to the study group, while women with regular monthly cycles with a biphasic thermal curve were recruited to the control group, as described in the 112-118 line. Exclusion criteria for control group: abnormalities in menstrual cycle regularity and symptoms that may indicate the presence of PCOS or other reproductive health disorders identified by medical history.

  1. Authors can alternative cite for the sub-sections from 2.1 to 2.6 sections in the methodology.

Subsections 2.1 to 2.6 have been supplemented with additional citations. The methodology described in these subsections focuses mainly on the assumptions of the InVivo method, which has been published in the following works:

Kicińska, A.M.; Stachowska, A.; Wierzba, T. Biowskaźniki Płodności, Obserwacja Wg Metody InVivo; Via Medica: Gdańsk, 2020; ISBN 978-83-66775-68-8.

Kicińska, A.M.; Stachowska, A.; Kajdy, A.; Wierzba, T.H.; Maksym, R.B. Successful Implementation of Menstrual Cycle Biomarkers in the Treatment of Infertility in Polycystic Ovary Syndrome—Case Report. Healthc. 2023, 11, doi:10.3390/healthcare11040616.

  1. Authors need to explain or give the reference for including the cases and controls or add the sample size formula.

We conducted a pilot study, and the sample size was determined based on available resources, such as the duration of the study and the availability of participants (the number of patients presenting to the Infertility Treatment Clinic during the study period and the number of healthy women who were willing to learn how to conduct cycle observations). We plan to continue our study to achieve greater representativeness in the future.

  1. BMI is very important element towards the PCOS women. If there is any probability, authors can add the BMI details in the revised manuscript.

All patients with PCOS and the control group did not fulfill the criteria for obesity, as described, most likely due to the fact that in such a young group of people typical metabolic disorders had not yet manifested themselves or were in the mildly symptomatic phase. However, we do not have their BMI value.

  1. Please do add the mean ages of both the groups of this studied women.

Due to the lack of a normal distribution as to the number of cards analyzed in both study groups, statistical analysis was carried out by performing the non-parametric U-Mann-Whintey test. Each woman and the data from her observation cards were treated as a separate case and the median of each parameter was calculated for them. In this circumstance, the mean will not represent the real central tendency, which makes its reporting less meaningful, and a better parameter to characterize the study groups will be the median.

  1. The majority of the results showed the negative association and it may be due to low sample size. Is there any alternative answer to justify the raised question.

A limitation of the study is the small size of the study group. The reason for the small number of cases analyzed is the need for an individual and extremely time-consuming process of training and accompanying the patient while conducting observations of the menstrual cycle.

  1. The quality of figure-3 is not maintaining the good and high resolution.

The quality of figure 3 (in the current version of the work Figure number 4) has been improved.

  1. Authors need to describe and explain in detailed towards using the FABMs method in previous studies and compare the current study results with the prior studies.

No study addressing the issue of menstrual cycle observation in a group of women with PCOS was found in the available literature, which demonstrates the innovation of our investigation. As a prelude to the study we created an InVivo observation card, which was published in a case report in the Healthcare. The example of a patient with PCOS described in the above-mentioned manuscript shows that reproductive success was achieved thanks to the multifactorial therapy used, but in relation to the individual, currently running cycle. Careful observation of the menstrual cycle and appropriate application of progesterone in relation to fertility bioindicators, according to the InVivo method [1]

The only method that allows for a standardized determination of the type of vaginal secretions is the single-index method according to Hilgers – the Creighton Model FertilityCare (CrSM) [2]. However, this method does not take into account the measurement of basal body temperature, therefore, among patients with PCOS, who often experience delayed ovulation and several mucus cycles preceding follicle rupture, it is impossible to determine without the use of ultrasound examination, whether a given mucus cycle was a cycle ending with ovulation or only follicle luteinisation (a phenomenon typical for PCOS). The combination of a standardized description of vaginal secretions with basal body temperature (BBT) measurement provides reliable help for patients with PCOS, as it allows them to determine when they actually ovulated, confirmed by an appropriate BBT spike. In addition, the number of photos posted in the InVivo gallery is more numerous and detailed. A specially created color InVivo card allows the patient to directly mark specific mucus features without having to learn complicated mucus definitions, as is the case in CrMS.

Summing up, the InVivo method is the first unique Fertility Awareness-Based Method, based on which a trained doctor can diagnose the condition of the reproductive tract and systemic diseases that may affect the quantity and quality of mucous secretions.

[1] Kicińska, A.M.; Stachowska, A.; Kajdy, A.; Wierzba, T.H.; Maksym, R.B. Successful Implementation of Menstrual Cycle Biomarkers in the Treatment of Infertility in Polycystic Ovary Syndrome—Case Report. Healthc. 2023, 11, doi:10.3390/healthcare11040616.

[2] Hilgers, T.W.; Stanford, J.B. Creighton-Model NaProEducation Technology for Avoiding Pregnancy. J. Reprod. Med. 1998, 495–502.

  1. Can authors explain the impact of this study in PCOS women.

Through their menstrual cycle observations, the participants learned about the cycle physiology and gained a greater awareness of their fertility. By understanding the fertility bio-indicators of a normal menstrual cycle, women could more easily recognize abnormalities occurring in their cycles, and observe improvements in cycle regularity during PCOS treatment. Analysis of the menstrual cycle can be a starting point for testing for e.g. metabolic diseases.

  1. Define pros and cons of this study.

The study had the advantage of parallel observation of basal body temperature and cervical mucus by the participating women and their regular contact with a menstrual cycle monitoring instructor. A significant benefit of the study was that all women photographed their cervical mucus and compared its appearance to the image of mucus presented in the pictorial dictionary, which minimized the possibility of error in noting mucus characteristics on the observation card. It also provided an opportunity to detect mucus with infectious characteristics and prompted further diagnostics.

A disadvantage of the study is the small size of the study groups and the lack of additional data, e.g., BMI values, fasting glucose and insulin levels, total cholesterol, LDL fraction cholesterol, HDL fraction cholesterol, triglyceride levels, androgen hormone levels, in order to draw attention that cycle disorders may be a prelude to metabolic disorders occurring in women with PCOS. On the other hand, the study enrolled young women in whom these metabolic disorders may not yet have appeared.

The aforementioned information has been added to the “discussion” section (lines: 652-664).

  1. Precise the conclusion.

The study demonstrated the usefulness of menstrual cycle observation using the InVivo method in healthy patients and patients with PCOS because it allows for objectification vaginal secretions examination.

Using a pictorial dictionary, whose macroscopic mucus photos correlate with the properties shown in the microscopic examination, provides evidence of the observations' correctness.

The in Vivo method allows the doctor to obtain a clinically reliable tool in the diagnosis and treatment of reproductive health disorders, including the supplementation of progesterone in accordance with the functioning of the ovary, in the second phase of the menstrual cycle.

The innovation consists in combining the description of vaginal secretions, according to the pictorial dictionary, with simultaneous measurement of basal body temperature, which gives women with abnormal cycles a better understanding of their course and may suggest to the doctor the need to diagnose other diseases, such as metabolic syndrome. 

Reviewer 4 Report

Comments and Suggestions for Authors

Comments to Authors 

            This study showed that systematic maintenance of monthly cycle observation charts using the InVivo method can be an important supplement to the medical history, as it allows for a thorough assessment of, among others, the timing of monthly bleeding, cervical mucus symptom, BBT changes, and the duration of the follicular and luteal phases among both healthy and PCOS women.

          Polycystic Ovary Syndrome (PCOS) is a prevalent endocrine disorder in women of reproductive age, affecting 5-15% globally with a large proportion undiagnosed [1]. The pathophysiology of PCOS is complex [2]. Recent studies have reported that apart from hyperandrogenism, insulin resistance, systemic chronic inflammation, and ovarian dysfunction, gut microbiota dysbiosis is also involved in PCOS development and may aggravate inflammation and metabolic dysfunction, forming a vicious cycle [2]. As naturally occurring plant secondary metabolites, polyphenols have been demonstrated to have anticancer, antibacterial, vasodilator, and analgesic properties, mechanistically creating putative bioactive, low-molecular-weight metabolites in the human gut [2].

          Authors are kindly requested to emphasize the current concepts about these issues in the context of recent knowledge and the available literature. This articles should be quoted in the References list.

References

1.      Implantation and Decidualization in PCOS: Unraveling the Complexities of Pregnancy. Int J Mol Sci. 2024; 25 (2): 1203. Published 2024 Jan 18. doi:10.3390/ijms25021203.

2.      Role of polyphenols in remodeling the host gut microbiota in polycystic ovary syndrome. J Ovarian Res. 2024; 17 (1): 69. Published 2024 Mar 27. doi:10.1186/s13048-024-01354-y.

Comments on the Quality of English Language

 Minor editing of English language required

Author Response

Response to Reviewers comments

Thank you for the opportunity to address the comments from the Reviewers. The authors hope that the Reviewers and Editors will be satisfied with the further amendments which we have made to the manuscript after taking on board the feedback.

Sincerely yours,

Aneta Stachowska

Department of Physiology

Medical University of Gdańsk

+48 58 349 1500

Dębinki 1, 80-211 Gdańsk

Title: Usefulness of the sympto-thermal method with standardized cervical mucus assessment (InVivo method) for evaluating the monthly cycle in women with Polycystic Ovary Syndrome (PCOS)

Reference: 2988683

Answers to the Reviewer 4

Dear Reviewer 4, we would like to thanks for taking effort necessary to review the manuscript. We sincerely valuable suggestions, which helped us to improve the quality of the manuscript. We agree with the reviewer that the manuscript should mention current concepts regarding PCOS in the context of the recent knowledge. Therefore, based on the two references cited, the introduction has been supplemented with the following information:

“Recent studies have shown that gut microbiota dysbiosis is also involved in the development of PCOS and may exacerbate inflammation and metabolic disorders. Polyphenols and their metabolites have been shown to have anticancer, antibacterial, vasodilatory and analgesic properties, which significantly alleviate systemic chronic inflammation occurring in women with PCOS [1].” (lines: 54-59)

“Up to 70% of cases of PCOS are undiagnosed, making it a significant concern for both patients and healthcare professionals [2].” (lines: 50-51)

  1. Role of polyphenols in remodeling the host gut microbiota in polycystic ovary syndrome. J Ovarian Res. 2024; 17 (1): 69. Published 2024 Mar 27. doi:10.1186/s13048-024-01354-y.
  2. Implantation and Decidualization in PCOS: Unraveling the Complexities of Pregnancy. Int J Mol Sci. 2024; 25 (2): 1203. Published 2024 Jan 18. doi:10.3390/ijms25021203.

Minor editing of English language required.

We have sent the Assistant Editor a certificate of translation into English.

Round 2

Reviewer 1 Report

Comments and Suggestions for Authors

The modification is well done.

Comments on the Quality of English Language

I cant see the language modifications.

Reviewer 2 Report

Comments and Suggestions for Authors

The article is very much improved. I agree with publication as it is now.